# Environmental and Socio-Demographic Influences on General Self-Efficacy in Norwegian Adolescents

**DOI:** 10.3390/bs15111484

**Published:** 2025-10-31

**Authors:** Catherine A. N. Lorentzen, Asle Bentsen, Elisabeth Gulløy, Kjell Ivar Øvergård

**Affiliations:** 1Department of Health, Social, and Welfare Studies, University of South-Eastern Norway, Pb 4, 3199 Borre, Norway; elisabeth.gulloy@usn.no (E.G.); kjell.oevergaard@usn.no (K.I.Ø.); 2The Competence Centre on Alcohol and Drugs—Region South (KORUS sør), Pb 1, Sentrum, 3701 Skien, Norway; asle.bentsen@korus-sor.no

**Keywords:** adolescents, Norwegian adolescents, general self-efficacy, environmental factors, socio-demographic factors, socio-ecological model

## Abstract

General self-efficacy is identified as a modifiable determinant of adolescent mental health and well-being. This study sought to better understand how conditions in different environments of adolescents’ lives and socio-demographic factors are associated with adolescents’ general self-efficacy. We conducted a hierarchical multi-variable linear regression analysis based on survey data from 2021 of a large population-based sample of Norwegian adolescents (*n* = 15,040). We found that better *Relation to peers* (*β* = 0.20, 95% *CI* [0.18; 0.22]) and *Academic/social relation to teachers* (*β* = 0.13, 95% *CI* [0.11; 0.14]), *Perceived neighbourhood safety* (*β* = 0.08, 95% *CI* [0.06; 0.10]), and *Participation in physical activities* (*β* = 0.07, 95% *CI* [0.06; 0.09]) had medium to small positive associations with adolescents’ general self-efficacy, whilst *Parental involvement*, *Participation in organized music/cultural leisure activities*, and *Perceived access to neighbourhood leisure arenas* had negligible associations with general self-efficacy. *Boys* reported a stronger general self-efficacy than *girls* (*β* = −0.17, 95% *CI* [−0.19; −0.16]) and *Age* and *Socio-economic status* had small positive associations with general self-efficacy (*β* = 0.08, 95% *CI* [0.07; 0.10] and 0.04, 95% *CI* [0.02; 0.06], respectively). We found some small moderation effects by socio-demographic factors in the associations between environmental factors and general self-efficacy. Our findings suggest that general self-efficacy-promoting initiatives that target adolescents apply a multi-sectorial and multi-level approach and pay particular attention to gender differences. A focus on facilitating adolescents’ experiences of mastery and access to relevant successful role models and supportive behaviour by adults and peers in the various contexts seems to be of particular importance.

## 1. Introduction

### 1.1. The Role of General Self-Efficacy for Adolescent Mental Health and Well-Being

In recent years, the mental health and well-being of adolescents in Norway and other Western countries has worsened ([17]; [61]; [70]; [82]). In an international meta-analysis, [82] ([82]) found that the pooled prevalence of elevated depressive symptoms among adolescents increased from 24% between 2001 and 2010 to 37% between 2011 and 2020. A large Norwegian study among adolescents showed an increase in psychological distress between the years 2014 and 2018 ([61]). Given these negative changes over time, a better understanding of modifiable determinants of negative and positive mental health outcomes is needed to inform health promotion initiatives.

General self-efficacy (GSE) has been identified as a potential modifiable predictor of mental health and well-being. In contrast to specific self-efficacy, which pertains to one’s confidence in handling abilities related to a particular task or domain (e.g., self-efficacy for physical activity), GSE refers to a general and stable sense of being capable of managing a wide range of stressful situations and everyday life challenges ([80]). Various studies in clinical and general populations of different ages show that general self-efficacious beliefs are negatively correlated with mental health problems ([12]; [24]; [39]; [52]; [76]) and positively correlated with subjective well-being and life satisfaction ([15]; [19]; [37]; [52]; [67]).

Theory and empirical research suggest that GSE affects mental health and well-being through multiple routes ([9]; [52]). The first is through increasing positive affect, which stems from having optimistic beliefs about one’s own competence and ability to master upcoming challenging situations ([52]). Second, GSE affects the person’s attitudes and actions, as people with high GSE are more likely to view challenging tasks and situations as something to be mastered, to select more ambitious goals, and to be persistent in pursuing their goals. This engagement and commitment in various activities is expected to lead to greater enjoyment in life. Those who doubt their capabilities tend to have lower levels of aspiration and commitment, avoid potential challenging situations and tasks, and will more easily experience stress ([9]; [69]; [80]). Third, those with a stronger GSE are more likely to adopt and engage in behaviours that may contribute to good mental health, such as physical activity and education ([11]; [52]). Thus, following the salutogenic model of health, GSE may be regarded as an individual cognitive general resistance resource that may be addressed in initiatives targeting adolescents’ mental health and well-being ([26]; [41]).

### 1.2. Theoretical Frameworks and Review of Literature Linking Environmental and Socio-Demographic Factors to GSE Among Adolescents

Theory and research highlight three key sources for the formation and strengthening of self-efficacy ([10]; [69]; [90]). The first, *direct personal experiences of mastery*, when success is attributed internally, provides insight into one’s capabilities and informs beliefs about the chance of succeeding in the future. Second, *vicarious self-efficacy experiences* involve social comparison processes that enhance one’s own self-efficacy when one observes social role models who successfully master difficult situations. Finally, *social persuasion* refers to convincing verbal persuasion about one’s own capabilities given by significant others ([10]; [69]; [90]). In addition to these sources, situated emotional and physiological states (e.g., excitement, fear, or fatigue) can also influence one’s self-efficacy beliefs ([10]).

In line with the socio-ecological perspective ([27]; [86], [87]) one could expect these sources of self-efficacy to exist in multiple layers or contexts in the adolescents’ environments. Socio-ecological models highlight the dynamic relationships between people and their surroundings and suggest that health-related outcomes at the individual level, such as GSE, for instance, are determined by various environmental factors in addition to personal factors. These environmental factors may include social relationships within the family, at school, and in leisure arenas, physical/structural aspects of the home, neighbourhood, and broader geographical surroundings, and political dimensions, such as local, regional, or national policies related to taxes, school class sizes, or leisure activity offerings. Health-influencing personal factors may include gender, age, and personality traits ([27]). Socio-ecological frameworks, which are at the core of health promotion approaches, emphasize the need to consider how health-related outcomes may be mutually influenced by personal factors and the multiple ecological systems surrounding the individual, and how factors in these different levels interact with each other ([27]; [86], [87]).

To date, only a limited number of studies have investigated whether and how factors in the environmental contexts of adolescents’ lives are linked to their GSE beliefs. Most of the existing studies have examined the influential role of the *home* environment. These show that family supportiveness ([29]; [54]; [81]; [88]), healthy family environments ([58]), nurturing parenting style ([39]), strength-based parenting style ([48]), and parental knowledge of ([29]) and interest in ([44]) their adolescents’ lives were positively associated with the adolescents’ GSE. On the other hand, indulgent parenting ([43]) and parental psychological control ([29]; [81]) were negatively linked to adolescents’ GSE. We have identified two studies that have examined the influencing role of *school* environmental factors. These revealed that enjoyment at school and good grades ([56]) and having caring teachers ([44]) were positively associated with GSE. A few studies have investigated the role of *peer* support, all of which support a positive relationship to GSE ([44]; [54]; [56]; [88]). There was a paucity of studies that investigated the impact of *leisure activities*. One study found that positive experiences from engagement in organized sport related positively to the adolescents’ GSE ([72]). Also, although organized through school, a martial arts-based ten-session intervention significantly increased total self-efficacy (composed of academic, social, and emotional self-efficacy) of secondary school students ([59]). Further, physical activity levels and GSE were positively correlated among both elementary school students ([31]; [94]), lower secondary school students ([31]), and undergraduate students ([98]). One study was identified that assessed the influencing role of the *neighbourhood* ([81]). This found that adolescents’ perceptions of neighbourhood assets such as safety, education, and wealth were positively related to their GSE.

This review of the empirical literature linking environmental aspects of adolescents’ lives to their GSE beliefs shows that this area of research is limited. Additionally, each existing study investigated the influencing role of only one or a few environmental factors, which prevents a more comprehensive understanding of the development of adolescents’ GSE. Further, the majority of the existing studies investigating the association between environmental characteristics and GSE among adolescents suffer from low statistical power due to small sample sizes (N for all but two studies between 100 and 843). Examining such associations in larger, population-based samples would provide a better basis for generalization.

With reference to a socio-ecological perspective, it is also relevant to address the socio-demographic characteristics of the adolescents when assessing contributors to their GSE. Indeed, several studies find higher GSE among boys than among girls ([11]; [40]; [44]; [50]) and among adolescents with higher socio-economic status ([11]; [16]; [44]; [55]). We found only one study that examined the role of age on GSE in the adolescent population, and this study found a slightly increasing GSE with higher grade levels among Norwegian lower secondary school students ([44]). Investigating the role of adolescents’ environments on GSE would be incomplete without controlling for these socio-demographic factors. Furthermore, assessing socio-demographic differences in GSE would inform whether there are subgroups in the adolescent population that need particular attention.

In line with the socio-ecological model, it would also be of interest to investigate whether the environmental and socio-demographic conditions of adolescents’ lives interact in the formation of their GSE. Such knowledge would point to whether some environmental predictors are particularly important to address in specific socio-demographic subgroups of the adolescent population. To our knowledge, there exist only two studies that have examined such interaction effects. The first study found a stronger positive association between family support and GSE for male than female adolescents; however, gender did not moderate the association between support of friends and GSE ([54]). The second study found no moderating effect of gender and grade level on the strength of the relationship between social support from friends, teachers, or parents, respectively, and GSE, among 13–16-year-old Norwegian adolescents ([44]). Thus, the scarcity of evidence and the lack of consistent empirical findings imply a need for further exploration of such interaction effects.

Summarized, this literature review underscores the necessity for more knowledge about the complex web of environmental and socio-demographic antecedents of adolescent GSE beliefs, including the interplay between these predictors, in larger, population-based adolescent samples.

### 1.3. Aims

This study aims to address identified shortcomings in the current understanding of the relationships between environmental and socio-demographic factors, respectively, and GSE among adolescents. Specifically, it seeks to: (1) use data from a large population-based sample of Norwegian adolescents to assess the association between five types of modifiable environmental conditions (home, school, peers, leisure activities, neighbourhood) and adolescents’ GSE; (2) control these associations for the impact of socio-demographic factors such as gender, age, and socio-economic status, as well as investigate socio-demographic differences in GSE; and (3) investigate whether these socio-demographic factors might have a moderating effect on the relation between environmental conditions and GSE.

Based on the reviewed literature, we hypothesized that perceptions of supportive home environments, supportive academic and social school environments, and positive peer relationships would be positively associated with adolescents’ GSE. Likewise, we anticipated that participation in organized leisure and physical activities, as well as positive perceptions of the neighborhood, would also show positive associations with adolescents’ GSE. Furthermore, we hypothesized that boys, older adolescents, and those from families with higher socio-economic status would report higher levels of GSE. Given the limited research on the moderating role of socio-demographic factors in the relationship between environmental conditions and adolescents’ GSE, these analyses were considered exploratory in nature.

## 2. Materials and Methods

### 2.1. Design

This study used a cross-sectional research design with data from the Norwegian national survey data collection scheme *Ungdata*. The Ungdata data collection scheme is designed to collect self-reported youth surveys at municipal level ([30]). The survey covers various aspects of young people’s lives that are relevant in a public health perspective, including mental health and well-being, social relations, school conditions, leisure time activities, health related behaviours, access to health-enhancing arenas, and personal resources. The Norwegian Social Research Institute, NOVA/OsloMet, coordinates the project in Norway ([65]), whilst the regional Drug and Alcohol Competence Centers (KORUS) ([45]) conduct the municipal surveys. The surveys are usually arranged in all municipalities within a given county at the same time.

### 2.2. Data Collection, Sample, and Study Context

The current study used already existing data from the 2021 Ungdata survey that was carried out in Vestfold and Telemark County in the south-eastern part of Norway (as of 2024, this county has been divided into Vestfold County and Telemark County). All students in secondary schools in the county’s 23 municipalities were invited to participate. Adolescents participated by responding to the online Ungdata questionnaire during school hours ([64]). In the Ungdata scheme, no efforts are made to collect data from adolescents who have dropped out of school or who are absent for the actual period of data collection. In total, the questionnaire was answered by 22,028 students, which represents a response rate of 86% in lower secondary schools and 72% in upper secondary schools ([1]). Those included in the current study were the 15,040 students who responded to all items included in the variables in the final multi-variable analysis. There was a higher proportion of girls (52% vs. 45.6%, *X*^2^ (1, *N* = 21,580) = 74.44, *p* < 0.001), a slightly higher age (*M* = 16.38, *SD* = 1.62 vs. *M* = 16.07, *SD* = 1.59, *M*_diff_ = −0.31, 95% *CI* of *M*_diff_ = [−0.35, −0.26]), and a slightly higher socio-economic status (*M* = 2.25, *SD* = 0.42 vs. *M* = 2.19, *SD* = 0.48, *M*_diff_ = −0.06, 95% *CI* of *M*_diff_ = [−0.07, −0.05]) among those included in the analyses versus those who were excluded due to missing data.

The data collection took place during the COVID-19 pandemic (spring 2021), following a year marked by multiple periods of lock-down, social restrictions, and limited access to key areas essential for adolescent development and well-being, including schools, arenas for organized and informal leisure activities, and health and welfare services ([34]).

### 2.3. Measures

#### 2.3.1. General Self-Efficacy

*GSE* was measured by the Norwegian short version of the General Perceived Self-Efficacy Scale ([74]; [79]). The original ten-item scale has been found to have good psychometric properties across many countries and cultures ([50]; [77]; [78]; [80]). The Norwegian short version used in the current study includes the following five statements: “I always manage to solve difficult problems if I try hard enough”, “I feel confident that I would be able to deal with unexpected events in an effective way”, “I remain calm when I face difficulties because I trust my ability to cope”, “If someone opposes me, I can find the means and ways to get what I want”, “If I’m in a predicament, I usually find a way out”. Responses were made on a four-point scale: 1 = Completely wrong, 2 = Quite wrong, 3 = Quite true, and 4 = Completely true. We calculated a mean score of all five items for each respondent. The final variable thus had a scale ranging from 1 to 4, with a higher score indicating a stronger GSE. This Norwegian short scale version has previously been found to be unidimensional and to have acceptable reliability, and the response categories were found to work quite well in a large population-based sample of Norwegian adolescents ([85]). Furthermore, this version showed good internal consistency in a large sample of Norwegian women ([97]) as well as in this study’s sample (α = 0.87).

#### 2.3.2. Environmental Variables Affecting GSE

Initially, twenty-seven items from the Ungdata survey were identified as relevant environmental predictors of adolescents’ GSE and were selected for the study. These items are presented in Table 1. The items reflected the following contexts: *home* (items 11–13), *school* (items 1–3, 6, 20–22), *peers* (items 4 and 5), *leisure activities* (items 14, 15, 18, 19, 23–27), and *neighbourhood* (items 7–10 and 16–17) (see Table 1).

We conducted preliminary exploratory factor analyses with these 27 items, entering them into a Maximum Likelihood Exploratory Factor Analysis (EFA) with oblique rotation (Direct Oblimin Rotation with delta = 0) and Eigenvalue > 1 as a selection criterium for inclusion of factors. Further criteria for selection of items were (1) items had to have a factor loading >0.30 on the main factor, (2) items had to be conceptually similar with other items loading on the same factor, and (3) in case of cross-loadings, the difference between highest and second highest factor loading had to be >0.25. If an item did not fulfil criteria 1–3, the item was removed and the analysis re-run.

A total of five items were excluded in this process (items 23–27 in Table 1). The final model contained seven factors with a total of 22 items. Table 1 shows the final rotated factor solution, containing the seven factors as well as the descriptive characteristics of the 22 items.

The first factor was interpreted to reflect the adolescents’ *Relation to peers*. The second factor reflected the adolescents’ *Perceived access to arenas for physical and social leisure activities in their neighbourhood*. The third factor assessed the adolescents’ perceptions of their *Parents’ involvement* in their life. The fourth factor assessed the adolescents’ *Participation in organized music/culture-related leisure activities*. The fifth factor reflected the adolescents’ *Feelings of safety in their neighbourhood*. The sixth factor reflected the adolescents’ *Participation in physical activities*. The last factor assessed the adolescents’ *Academic and social relation to their teachers*. These factors constituted the final seven environmental independent variables. Because included items had somewhat different scales, we saved the standardized regression factor scores for later use as indicators of the underlying environmental constructs in regression analysis.

#### 2.3.3. Socio-Demographic Variables

*Gender* was measured with response options “Boy” and “Girl”. Grade level was used as a proxy for *Age* (e.g., those in the first year of secondary school = 14 years, those in the third year of upper secondary school = 19 years). *Socio-economic status* was assessed by combining answers to four questions from the Family Affluence Scale II (the family having a car, computers, and travelled on holiday, and the adolescent having his/her own bedroom) ([23]), one question about the parents’ educational level, and one question about the number of books in the home. In line with instructions from the developers of the questionnaire, response options for each of these six questions were coded to represent a scale ranging from 0 (low socio-economic status) to 3 (high socio-economic status) ([6]). For example, the parental educational scale included: None of the parents have higher education = 0, One of the parents = 1.5, Both parents = 3. A mean score of these six scales was then calculated.

#### 2.3.4. Procedure for Ensuring Validity of Ungdata Measures

The Ungdata questionnaire contains items that have been developed specifically for the Ungdata study and items that have been used in previous studies—mostly studies among Norwegian adolescents ([30]). The Ungdata questionnaire is revised every three years to ensure that the items remain relevant and function as intended for new cohorts of adolescents. The revision process is conducted in collaboration with research and expertise networks, relevant public authorities, and the adolescents themselves.

### 2.4. Analyses

All statistical analyses were performed with IBM SPSS 28. Characteristics of the sample are presented through descriptive analyses of the dependent, the environmental, and the socio-demographic variables; mean and standard deviation (SD) for continuous variables, and sub-sample size (*n*) and percentage for categorical variables. Due to differing scales for items included in one of the environmental variables, the mean and SD of environmental variables were based on sum scores of the original items.

The Pearson linear correlation coefficient (*r*) was used for correlations between continuous variables, whilst correlations between the dichotomous variable gender and the other variables were assessed through point-biserial (*r*_pb_) correlation analyses.

Hierarchical multi-variable linear regression with three steps was used to investigate the unique contribution of each of the covariates. We applied a hierarchical model to assess how the coefficients changed when adding new groups of covariates. In Step 1, the seven environmental constructs from the EFA were included as independent variables (X_1_, X_2_, … X_7_) for the prediction of the dependent variable GSE (Y). Step 2 added the socio-demographic variables gender, age, and socio-economic status (X_8_, X_9_, X_10_), and Step 3 included the 21 interaction terms (X_11_–X_31_) of the seven environmental constructs multiplied by the three socio-demographic variables. Statistically significant interaction effects were further explored using Model 1 in the PROCESS application in SPSS 28 ([38]). In these analyses, GSE was entered as the dependent variable (Y), the actual environmental variable was entered as the independent variable (X), the actual socio-demographic variable was entered as the moderator (W), and all other independent environmental variables, socio-demographic variables, and interaction terms from the original hierarchical regression Model 3 were entered as covariates. Simple slopes analyses were conducted to aid in the interpretation of these interaction effects. Assumptions for multi-variable linear regression analysis (non-collinearity, normality, homoscedasticity, and linearity) were tested and found to be met ([28]). Missing data was handled by listwise deletion ([38]).

### 2.5. Ethics

This study used already existing anonymized data provided by NOVA. The data collection, carried out by NOVA and KORUS, was conducted with informed consent, including a passive consent from guardians of students under the age of 18 ([66]), and was approved by the Norwegian Agency for Shared Services in Education and Research (Project title: Ungdata 2020–2022, project number 821474).

## 3. Results

### 3.1. Descriptive Statistics

Table 2 shows descriptive data for GSE, the seven environmental predictors, and the three socio-demographic variables.

The correlation matrix for GSE, the seven environmental factors, and the three socio-demographic variables can be seen in Table 3.

### 3.2. Hierarchical Multi-Variable Linear Regression Analysis

Results from the hierarchical multi-variable linear regression analysis with three steps are shown in Table 4. In the first step, Relation to peers (*β* = 0.21, 95% *CI* [0.19, 0.23]), Perceived safety in the neighbourhood (*β* = 0.14, 95% *CI* [0.12, 0.16]), and Academic and social relation to teachers (*β* = 0.14, 95% *CI* [0.12, 0.16]) all had medium to small positive associations with GSE, which indicates stronger GSE among adolescents with more and better relationships to peers, who are more inclined to perceive their neighbourhood as safe, and who are more inclined to feel cared for by their teachers and understand their teaching and feedback. Participation in physical activities had a small positive association with GSE (*β* = 0.08, 95% *CI* [0.07, 0.10]), which indicates a somewhat stronger GSE among adolescents who are more physically active. Parental involvement, Participation in organized music/cultural leisure activities and Perceived access to arenas for physical and social leisure activities in the neighbourhood had negligible associations with adolescents’ GSE. In total, the predictors explained 16.4% of the variance in GSE.

Inclusion of the socio-demographic factors in Step 2 did not substantially change the associations of the environmental predictors from Step 1. Step 2 revealed that *Gender* had a medium to small negative association with GSE, which reflects a stronger GSE among boys than among girls (*β* = −0.17, 95% *CI* [−0.19, −0.16]). *Age* and *Socio-economic status* had small positive associations with GSE (*β =* 0.08, 95% *CI* [0.07, 0.10] and 0.04, 95% *CI* [0.02, 0.06], respectively), suggesting a somewhat stronger GSE among older adolescents and those from families with greater socio-economic resources. Total explained variance in GSE increased to 19.6% in Step 2.

Step 3 added the 21 interaction terms between the environmental predictors and the socio-demographic variables. The analysis showed that *Gender* (coded as boys = 1, girls = 2) moderated the relation between *Relation to peers* and GSE, with girls having a slightly larger positive association than boys (interaction: *β* = 0.07, 95% *CI* [0.01, 0.13]) (see Figure 1a), and between *Academic and social relation to teachers* and GSE (girls again having a slightly larger positive association than boys, *β* = 0.11, 95% *CI* [0.06, 0.17]) (see Figure 1b). *Gender* also moderated the association between *Perceived safety in the neighbourhood* and GSE, which was indicated by a somewhat higher positive association among boys than among girls (interaction: *β* = −0.10, 95% *CI* [−0.16, −0.04]) (see Figure 1c).

Step 3 also shows an interaction effect between *Age* and *Parental involvement* on GSE (interaction: *β* = −0.10, 95% *CI* [−0.14, −0.06]), which shows a small positive association between *Parental involvement* and GSE for the youngest participants, and an increasingly negative association for the older age groups, with the average association shifting from positive to negative between the ages of 15.7 and 15.9 (see the interaction effect in Figure 1d). No other interaction effects were detected.

## 4. Discussion

The aims of this study were to examine how conditions in different environments of adolescents’ lives and socio-demographic factors were associated with their GSE. We also wanted to assess the moderating effects of socio-demographic factors on the association between environmental conditions and GSE. The findings show that GSE was mainly associated with having good relationships with peers within and outside school, by perceiving that the teachers are caring and that their teaching and feedback is understandable, by perceiving the neighbourhood as safe, and by participation in organized and informal physical activities. Furthermore, boys reported higher GSE than girls, and GSE increased slightly with age and with socio-economic status. Interaction analyses revealed that girls had slightly higher positive associations between GSE and relations to peers and teachers than boys, whilst the positive association between GSE and perceptions of living in a safe neighbourhood was somewhat higher for boys than for girls. Further, there was a positive association between parental involvement and GSE among the youngest adolescents, but this association was negative among the oldest.

These findings contribute with new knowledge as no previous studies have simultaneously included factors that cover most of the contexts in which adolescents live their lives when analyzing influences on GSE. Our findings indicate that adolescents’ GSE is multi-determined, by experiences stemming from various environments, which is in line with previous research ([43]; [44]; [54]; [56]; [59]; [72]; [81]; [88]; [94]). Our findings also align with the socio-ecological approach in health promotion, which highlights the interplay between people and their surroundings in the formation of health and the necessity to consider various levels of influence, such as social, physical, structural, and political in addition to personal, when designing health-promoting interventions ([27]; [86], [87]). Accordingly, our findings support the need to simultaneously address determinants in various sectors and at different levels to more effectively obtain strengthened GSE beliefs among adolescents.

The two environmental factors most strongly associated with adolescents’ GSE in our study were peer and teacher relationships. The observed link between positive peer relationships—both within and outside the school setting—and adolescents’ GSE resonates with prior research that highlighted the significant role of peer support ([44]; [54]; [56]; [88]). Based on theory and previous empirical studies ([10]; [69]; [90]), it is likely that adolescents with stronger positive peer relationships build higher GSE through increased experiences of mastery (e.g., developing social competencies), exposure to successful role models, and encouragement from peers who reinforce confidence in their abilities. Additionally, positive peer relationships may contribute to enhanced emotional and physiological states, such as reduced stress and increased feelings of belonging, which are also expected to strengthen GSE ([10]).

Similarly, the positive link discovered between adolescents’ perception of their teachers as caring and comprehensible in their instructions and feedback and their GSE aligns with the findings of previous studies that examined the role of school environmental factors ([44]; [56]). For instance, a longitudinal study among 768 US adolescents found that school success, in terms of enjoyment of school, perceived ability to focus on school tasks, and school grades, was positively related to GSE ([56]). It is plausible that adolescents who perceive their teachers as caring feel safer and more supported in the learning environment. This sense of safety, combined with clear and comprehensible instructions and feedback, may foster more opportunities for academic-related mastery experiences. Additionally, perceptions of having caring teachers may reflect the presence of teachers who provide encouragement and instill confidence in students about their abilities. Together, these elements are likely to contribute to strengthened GSE ([10]; [69]; [90]). These findings suggest that strategies that focus on the school and leisure contexts, including teaching and leadership approach and peer relationships, may be particularly influential on adolescents’ general beliefs in their ability to cope with upcoming challenging events ([42]; [44]; [54]; [56]; [72]; [88]).

Regarding the family context, our results revealed no influential role of parental involvement in the adolescents’ lives on their GSE beliefs across the entire sample. This is in contrast to most previous studies with samples with a similar age range that address the role of different types of family supportive factors in the formation of adolescents’ general self-efficacy ([29]; [54]; [81]). Such a discrepancy may be related to the use of different measures; we included items assessing adolescents’ perceptions of their parents’ knowledge about their leisure time and friends and interest in their life, whilst most other studies assessed adolescents’ perceived support from parents and family. However, our interaction analyses revealed that parental involvement was positively associated with GSE beliefs among the younger participants, but that this association weakened with higher age, and became slightly negative among the older adolescents. This finding aligns with the results of [44] ([44]), who, similarly to us, assessed parental support through adolescents’ perceptions of their parents’ interest in their life, and who included only younger adolescents in their study. The observed pattern in this study can be interpreted through developmental theories of attachment and autonomy. While psychological theories vary in their emphasis on distinct versus gradual developmental phases ([20]), reorientation away from parents and family life towards peers, partners, and social life outside the home is considered a key process of adolescence ([95]). The need for autonomy gradually replaces the need for attachment and security provided by parents. Moreover, according to self-determination theory, autonomy, structure, and involvement are critical elements of a healthy family environment for adolescents, while controlling parental behaviors are likely to have negative effects ([75]). Thus, and as discussed by various researchers ([29]; [81]), younger adolescents who perceive that their parents are interested in and knowledgeable about their lives, activities, and friends may feel supported and validated. This sense of support may foster feelings of positive self-esteem to handle upcoming difficult situations. In contrast, as older adolescents seek greater independence and autonomy, similar parenting practices may be perceived as controlling or as a lack of trust in their capabilities. This may have a counteracting influence on the formation of their GSE beliefs.

In line with previous studies on the role of physical activity ([31]; [94]; [98]) and sporting participation ([59]; [72]), our findings indicate that enhanced physical activity levels may also contribute to strengthened GSE beliefs among adolescents. Although scarcely researched, plausible explanations for this association may involve bio-psychological mechanisms. For instance, perceptions of increased muscular and cardiorespiratory fitness that result from physical activity can act as a mastery experience, which may generalize to a broader physical self-concept, and consequently to enhanced GSE ([21]; [62]). Furthermore, the reduction in hormonal and metabolic stressor reactivity of the hypothalamic–pituitary–adrenal axis through regular physical activity may alter the way in which one interprets, reacts to, and recovers from stressful situations, which may further influence general capability beliefs ([91]). Regardless of the mechanisms underlying this association, our findings support a well-documented need for a continued focus on initiatives to increase physical activity levels among adolescents ([93]).

Perceptions of one’s neighbourhood, such as feeling safe and perceiving having access to arenas for recreational activities, would be expected to promote a general sense of capability ([32]; [81]). This could occur directly, through perceptions of meeting lesser obstacles in life and more resources to overcome potential challenges, and indirectly, by the provision of more opportunities for mastery experiences, vicarious learning, and social support through more engagement in social and physical activities. In congruence with results from a US-based study ([81]), we found perceptions of safety in the neighbourhood to be of significance for adolescents’ GSE. However, contrary to what one could expect, this was somewhat more important for boys’ self-efficacy beliefs than for girls’. One possible explanation may be found in the different gender roles that are related to controlling one’s environment. Across nations and cultures, including in this study, females report lower levels of perceived safety in their neighbourhoods compared to males ([13]; [25]). This may be related to a higher risk of harassment among females, but also to lesser physical capabilities to be in control of a physical confrontation ([13]). Thus, although being more vulnerable, girls might have lower expectations of being in control of their neighbourhoods than boys, which may translate into a lower association between perceptions of safety and GSE. The observed interaction effect may also be related to differences in neighbourhood exposure patterns between genders. Research has shown that boys are more likely than girls to have local social networks and spend more time in the local environment, particularly as they grow older and their radius of activity expands ([14]). Furthermore, boys tend to experience fewer mobility regulations and other parental strategies aimed at reducing potential negative effects from neighbourhoods perceived as dangerous or unfortunate by parents ([83]). Consequently, boys are likely to be more exposed to both positive and negative neighbourhood experiences. Aligning this, studies from various parts of the world indicate that boys are more influenced than girls by neighbourhood characteristics—both positively (e.g., social cohesion) and negatively (e.g., exposure to violence, poverty, or education-hostile environments) ([14]; [63]; [71]). Boys from disadvantaged families or boys living in impoverished neighbourhoods are particularly vulnerable to these influences. These factors may help explain why perceived neighbourhood safety is more strongly associated with boys’ GSE beliefs compared to girls’.

Regarding socio-demographic influences on GSE, we found that girls expressed less faith than boys in their general ability to tackle upcoming challenges. This finding is comparable with the results of several previous studies ([11]; [44]; [50]; [68]) and aligns with the poorer mental health observed among adolescent girls in both Norwegian ([46]; [61]) and international studies ([82]). This gender difference may be linked to disparities in perceived pressure and stress. For instance, a Norwegian study on different types of pressure found that girls reported higher levels of school-related and body-related pressure, with social media use playing a mediating role in the latter ([7]). Aligning this, girls generally report higher levels of perceived stress compared to boys ([7]; [35]; [49]), which may reflect fewer opportunities for mastery experiences. Explanations for these gender effects span a range of theoretical frameworks, from psychological theories of stress exposure and vulnerability/reactivity ([51]; [60]) to gender theories of power ([3]; [73]) and broader social theories of individualization and “the schooled society” ([5]; [33]). The interplay of educative mindsets, optimization strategies, and digital algorithms may disproportionately affect girls, exacerbating their vulnerability. These findings highlight a gendered vulnerability within the adolescent population and emphasize the importance of applying a gender-sensitive approach in GSE-promoting work. While the literature points to a complex web of influences requiring multi-level and multi-faceted strategies, our interaction effect findings suggest that interventions focusing on teacher and peer relationships may be particularly beneficial for girls.

The small positive association between age and GSE observed in this sample is similar to that of other studies ([44]), and might reflect both that older adolescents have had more opportunities for mastery experiences, vicarious learning, and supportive verbal persuasion ([10]), and that they develop a stronger sense of capability and control over their surroundings through the natural maturational process of becoming detached and independent from their parents and other caregivers.

Furthermore, we found a small positive contribution of family socio-economic status to the adolescents’ GSE. This aligns with previous studies ([11]; [16]; [44]; [55]) and can be explained through various mechanisms. For instance, some researchers ([55]; [81]) highlight how adolescents from families with higher socio-economic status may have better living conditions and more resources that support mastery experiences in several fields. They may also be more often exposed to positive role models in various arenas—for instance the passing on of skills that are useful in the educational system. Although reflecting small effects, these results still support a continued focus upon broad social determinants of health within a socio-ecological model, such as family welfare, education, and employment, in order to optimize the work of strengthening adolescents’ capability beliefs.

### 4.1. Practical Implications

Taken together, the knowledge that has emerged through this study may help inform initiatives to enhance adolescents’ general beliefs in their capabilities to tackle difficult circumstances, which could contribute to the slowing down of the negative development in young people’s mental health that has been observed in the past decade ([17]; [61]; [82]). First and foremost, for optimizing this work, our findings suggest that simultaneously targeting multiple contexts in which adolescents live their everyday lives, including at school, in leisure time, at home, and in the neighbourhood, may be valuable. The influential roles of peers, teachers, leisure activity leaders, and parents might be particularly important in this matter. As previously discussed, the application of a socio-ecological approach in the design of interventions and measures aimed at promoting adolescents’ GSE could be promising ([27]; [86], [87]). This may involve policy changes that aim to educate adults involved with adolescents in these various contexts about the pivotal role of adolescents’ GSE beliefs for their mental health and well-being, about variations in such beliefs among subgroups of the adolescent population, as well as about how such beliefs can be formed and strengthened. Theory and prior empirical evidence suggest that the latter may involve an educational focus on how to facilitate the support of key sources of self-efficacy—personal experiences of mastery, vicarious experiences of mastery, and social persuasion ([10]; [92]). This could, for instance, include a focus on designing tasks that are achievable with effort, tailored to each adolescent’s individual need and competence level; on highlighting their personal progress instead of focusing on normative goals or comparing with others; on reminding them that challenges are normal and helping them to frame mistakes as opportunities to learn and improve; on allowing them to navigate and address the challenges they encounter before offering assistance; on facilitating their perceptions of feeling useful, such as having them contribute to tasks or helping others—for instance, by encouraging them to help class or team mates to learn new skills; on providing role models who struggle but succeed, preferably from diverse identities and backgrounds so that different adolescents can have role models with whom they can identify; on establishing supportive and encouraging cultures and structures; and on ensuring that persuasive messages are sincere and specific ([10]; [18]; [42]; [43]; [44]; [92]). Such strategies could be included in teacher training programmes, leisure activity leader programmes, and parenting programmes. Educational programmes with similar learning content that target adolescents directly could be incorporated into the school curriculum. In terms of parenting style and engagement in their adolescents’ lives, our findings suggest the importance of striking a delicate balance. Parents might consider maintaining an involvement level that fosters autonomy and avoids creating feelings of control and mistrust in their older adolescents’ ability to navigate challenging situations ([43]).

Our findings additionally suggest that social and/or structural measures in the local communities that ensure neighbourhood safety among adolescents may help to improve GSE beliefs, particularly among boys. As part of their local public health work, municipalities could involve local adolescents in the identification of factors that counteract feelings of neighbourhood security and possible solutions to them. The results of this study also suggest a continued focus on reducing drop-out from sports leisure activities ([4]) and maintaining and increasing physical activity levels ([93]) among adolescents to ensure high levels of GSE beliefs. In this regard, research and theory underscore the need to ensure the fulfilment of the basic psychological needs of autonomy, competency, and relatedness in these activities ([4]; [18]).

Finally, the findings from the present study underscore the need to focus on increasing girls’ GSE beliefs.

### 4.2. Strengths and Limitations of the Study and Future Directions

As has been shown, our study adds to the knowledge base for mental health promotion work among adolescents by simultaneously investigating modifiable predictors of GSE from various contexts of their lives, and by examining the moderating role of socio-demographic factors in these relationships, in a large population-based sample of adolescents. However, there are several limitations to the study, which should be considered when interpreting the results.

First, the cross-sectional design of the study implies that we cannot make inferences about causal relationships and the directionality of effects. There is likely to be a reciprocal relationship between the environmental factors and GSE ([10]). For instance, adolescents with a stronger general faith in their coping skills may be more likely to seek sports clubs or peer social settings and may generally be better equipped to create or perceive supportive environments compared to those with less faith in their capabilities. Also, within a broader framework of adolescent well-being, some of the environmental factors included in the study, such as relationships with parents, peers, and teachers, may be understood as indicators of well-being or quality of life rather than predictors ([57]). This raises the possibility of confounding or overlapping constructs. To better establish causality, and clarify the direction of effects, future studies should apply longitudinal research designs.

Second, the independent variables included in our model accounted for merely 20% of the variance in adolescents’ GSE, which suggests the presence of other influential factors that were not considered in our model. For instance, including support from parents, siblings, and other family members, sleep duration ([36]), and ethnicity ([69]) in future research may offer additional insight into the development of GSE among adolescents. Moreover, resilience research highlights the potential importance of individual factors such as emotion regulation ([47]) and hardiness ([2]; [53]) in the formation of GSE, which also merits further exploration in more comprehensive models. Previous research also indicates a genetic component to GSE ([96]), thus challenging assumptions that beliefs of capability are primarily formed through mastery experiences and influences from the environment.

Third, the data collection for this study was conducted during the COVID-19 pandemic, following several periods of lock-down and social restriction measures. This may have influenced the answers for several questions related to both the dependent variable (GSE) and independent variables—for instance, lower participation in organized leisure activities and less contact with peers and teachers. Supporting this, a study using data from approximately 600,000 Norwegian adolescents collected between 2014 and 2022 found that participation in sports clubs and other organized leisure activities decreased somewhat during the pandemic compared to previous years ([8]). While pandemic-related effects likely influenced several variables, we do not have indications that this would change the associations between them. On the contrary, the unique context of the pandemic might even strengthen certain findings. For instance, the fact that peer relationships remained a significant predictor of GSE during a time of social restrictions may indicate particularly robust associations. Future studies could compare results from pandemic and non-pandemic periods to determine whether associations remain consistent across different contexts.

Fourth, the analytic sample differed somewhat from those excluded from the analyses due to missing data, in terms of gender proportion, age, and socio-economic status, which may introduce bias into the regression estimates. To address this, a sensitivity analysis was conducted using an imputed dataset generated with the Expectation-Maximization algorithm ([22]) (missing data for gender were not imputed, as this method does not support imputation of categorical variables). The regression coefficients from the imputed dataset were highly similar to those from the original analysis, indicating that the conclusions of this study were not affected by the missing data (see Appendix A). However, it is plausible that adolescents who were absent during the data collection period or who had dropped out of school differed from participants on relevant aspects of the study (e.g., GSE, socio-economic status). This may contribute to uncertainty when generalizing the results to the wider adolescent population of Vestfold and Telemark counties. These counties also have the highest proportion of young people classified as NEET (not in education, employment, or training) in Norway ([84]), potentially further complicating generalizability to the broader Norwegian adolescent population. Moreover, specific cultural or environmental differences should be accounted for when generalizing the findings to other countries’ adolescent populations. For example, feeling of safety in one’s neighbourhood may be more influential on adolescents’ GSE in countries that have higher crime and sexual harassment rates compared to Norway.

Finally, although our findings point to relevant predictors of GSE to be addressed in mental health-promoting work among adolescents, they do not inform well on how best to succeed in obtaining this in various contexts. Although the literature highlights sources for enhanced self-efficacy to target in interventions, there is a need for longitudinal and experimental research to identify which specific GSE-promoting initiatives that may be the most promising in these contexts—for example, in sports club contexts, in school contexts, and in family environments.

## 5. Conclusions

Based on survey data from a large population-based sample of adolescents from a Norwegian county, we assessed whether and to what degree aspects of adolescents’ environment and their socio-demographic characteristics were associated with their general belief in their capabilities to tackle challenging situations, and whether socio-demographic characteristics moderated associations between environmental factors and GSE. In line with the socio-ecological model, hierarchical multi-variable linear regression analyses indicated that adolescents’ GSE was influenced by both environmental conditions and socio-demographic aspects, and that some of these factors interacted in their association with GSE. More specifically, having positive peer relationships and supportive teachers whose teaching was understandable was associated with higher GSE, although these associations appeared to be somewhat stronger for girls than for boys. Participation in physical activities was positively associated with GSE among all adolescents, whilst a perceived safe neighbourhood showed a slightly stronger positive association with boys’ GSE beliefs compared to girls’. Boys reported higher GSE beliefs than girls, and both age and socio-economic status showed a slight positive correlation with GSE. These results advocate for a multi-sectoral, multi-level, and gender-sensitive approach in designing GSE-promoting initiatives as part of public health efforts.

## Figures and Tables

**Figure 1 behavsci-15-01484-f001:**
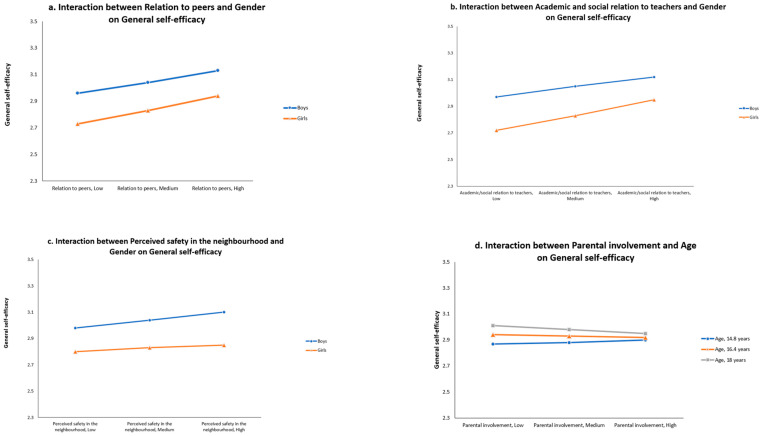
(**a**–**d**) Significant interactions between socio-demographic variables and environmental variables in their effect on general self-efficacy. Scale General self-efficacy = 1–4.

**Table 1 behavsci-15-01484-t001:** Final Rotated Factor Loading Plot from Exploratory Factor Analysis and Description of Included and Excluded Environmental Predictor Items (*n* = 15,040).

			Pattern Matrix—Factors						
	Original Scale	M (*SD*)	1	2	3	4	5	6	7
Included items									
1. I feel that I fit in with the students at my school	1–4 ^a^	3.20 (0.85)	0.70	0.00	−0.02	−0.02	0.02	0.03	0.11
2. Do you have someone to hang out with during breaks at school?	1–4 ^b^	3.70 (0.54)	0.66	0.02	0.02	0.01	0.02	0.00	−0.08
3. I enjoy school	1–4 ^a^	3.37 (0.76)	0.60	−0.01	0.02	0.01	−0.04	0.00	0.31
4. Do you have someone to hang out with in your spare time?	1–4 ^b^	3.44 (0.64)	0.57	0.04	0.03	−0.02	0.05	0.01	−0.10
5. Do you have at least one friend who you trust completely and who you can tell absolutely anything?	1–4 ^c^	3.54 (0.71)	0.45	0.03	0.05	−0.01	0.01	0.00	−0.05
6. I often don’t want to go to school	1–4 ^a^	3.18 (0.94)	0.44	−0.04	−0.05	0.02	0.04	0.05	0.23
7. Perceived access to culture venues in the local environment (cinemas, concert venues, libraries, etc.)	1–5 ^d^	3.51 (1.25)	0.01	0.79	−0.01	0.02	−0.02	−0.05	−0.02
8. Perceived access to sports facilities in the local environment.	1–5 ^d^	3.91 (1.09)	0.03	0.67	0.02	−0.01	0.05	0.18	−0.02
9. Perceived access to public transport in the local environment (buses, trains, trams, etc.)	1–5 ^d^	3.64 (1.21)	0.02	0.63	0.00	−0.02	−0.04	−0.04	0.03
10. Perceived access to places for meeting other young people in your free time in the local environment	1–5 ^d^	3.05 (1.30)	−0.04	0.62	0.00	0.01	0.04	−0.01	0.05
11. My parents usually know where I am, and who I’m with, in my free time	1–4 ^e^	3.62 (0.60)	−0.04	−0.01	0.77	−0.01	−0.03	−0.03	0.02
12. My parents know most of the friends I hang out with in my free time	1–4 ^e^	3.49 (0.70)	−0.01	−0.01	0.73	0.00	0.00	0.01	−0.02
13. My parents are very interested in my life	1–4 ^e^	3.45 (0.69)	0.09	0.02	0.49	0.00	0.06	0.04	0.05
14. How many times during the last month have you participated in activities with a culture/music school?	1–4 ^f^	1.16 (0.60)	0.00	0.00	0.01	0.81	0.01	0.01	−0.02
15. How many times during the last month have you participated in organized activities with a band/choir/orchestra?	1–4 ^f^	1.09 (0.44)	−0.01	0.00	−0.01	0.50	−0.01	0.00	0.01
16. How safe do you feel when you are out in the area where you live?	1–4 ^g^	3.55 (0.63)	0.01	−0.01	0.01	−0.01	0.81	−0.01	0.00
17. When you are out in the evening in your local area, do you feel safe?	1–4 ^h^	3.33 (0.78)	−0.01	0.00	−0.01	0.00	0.74	−0.01	0.02
18. How many times during the last month have you participated in activities in a sports club?	1–4 ^f^	2.15 (1.34)	−0.03	0.03	0.01	0.01	−0.03	0.75	−0.01
19. On how many days (the last seven days) were you so physically active that you were short of breath or sweaty for at least 60 min in total in one day?	1–5 ^i^	2.55 (1.12)	0.03	−0.02	0.00	−0.02	0.01	0.62	0.00
20. The feedback I get from the teachers is usually understandable	1–4 ^e^	2.96 (0.71)	−0.07	0.05	0.03	−0.02	0.03	−0.02	0.72
21. I usually understand well what the teachers mean when they explain something in class	1–4 ^e^	2.95 (0.73)	−0.01	0.02	0.03	0.02	0.07	0.05	0.61
22. My teachers care about me	1–4 ^e^	3.18 (0.76)	0.11	0.05	0.06	0.01	0.00	−0.04	0.56
Excluded items									
23. How many times (the last 7 seven days) have you spent most of the evening socializing on the Internet or by mobile phone?	1–4 ^j^								
24. How many times (the last 7 seven days) have you spent most of the evening playing online games?	1–4 ^j^								
25. How many times during the last month have you participated in after-school club/youth centre/youth club?	1–4 ^f^								
26. How many times during the last month have you participated in meetings with a religious organization?	1–4 ^f^								
27. How many times during the last month have you participated in another organization, society or association?	1–4 ^f^								

Note: ^a^ = 1 (Totally agree)–4 (Totally disagree); ^b^ = 1 (Yes, always)–4 (No, never); ^c^ =1 (Yes, definitely)–4 (There’s nobody I would call a friend at the moment); ^d^ =1 (Very good)–5 (Very bad); ^e^ = 1 (Fits very well)–4 (Fits very badly); ^f^ = 1 (Never)–4 (5 or more times); ^g^ = 1 (Very safe)–4 (Not safe at all); ^h^ = 1 (Yes, very safe)–4 (No, I don’t feel safe); ^i^ = 1 (None)–5 (7 days); ^j^ = 1 (None)–4 (6 or more times). When necessary, items were reverse-coded so that higher mean scores represent more positive values (better relationship, more access, more involvement, more participation, feeling safer, better understanding). Extraction method in the explorative factor analysis: Maximum Likelihood; Rotation method: Oblimin with Kaiser Normalization. Rotation converged in 8 iterations. Total explained variance = 46.42%.

**Table 2 behavsci-15-01484-t002:** Descriptive Data for the Dependent, Environmental, and Socio-Demographic Variables (*n* = 15,040).

Variable	Scale	*M* (*SD*)	*n* (%)
**Dependent variable**			
General self-efficacy	1–4	2.93 (0.61)	
**Environmental variables (sum scores of included items)**			
Parental involvement	3–12	10.56 (1.58)	
Relation to peers	6–24	20.44 (3.05)	
Academic and social relation to teachers	3–12	9.09 (1.75)	
Participation in physical activities	2–9	4.69 (2.10)	
Participation in organized music/cultural leisure activities	2–8	2.25 (0.87)	
Perceived safety in the neighbourhood	2–8	6.87 (1.27)	
Perceived access to arenas for physical/social activities in the neighbourhood	4–20	14.11 (3.74)	
**Socio-demographic variables**			
Gender			
Boys			7219 (48)
Girls			7821 (52)
Age		16.38 (1.62)	
14 years			2480 (16.5)
15 years			2639 (17.5)
16 years			2742 (18.2)
17 years			2873 (19.1)
18 years			2536 (16.9)
19 years			1770 (11.8)
Socio-economic status	0–3	2.25 (0.42)	

Note: Higher scores on scales represent more positive values (higher general self-efficacy, better relationships, more participation, feeling safer, more access, higher socio-economic status).

**Table 3 behavsci-15-01484-t003:** Correlation Matrix for the Dependent, Environmental, and Socio-Demographic Variables (*n* = 15,040).

Variable	1	2	3	4	5	6	7	8	9	10
1. General self-efficacy										
2. Parental involvement	0.17									
3. Relation to peers	0.36	0.42								
4. Academic and social relation to teachers	0.29	0.31	0.47							
5. Participation in physical activities	0.19	0.30	0.35	0.09						
6. Participation in organized music/cultural leisure activities	0.01	−0.02	−0.06	0.08	−0.08					
7. Perceived safety in the neighbourhood	0.31	0.29	0.51	0.40	0.21	−0.06				
8. Perceived access to arenas for physical/social act. in the neighbourhood	0.13	0.26	0.30	0.35	0.21	0.06	0.21			
9. Gender	−0.23	0.08	−0.11	−0.14	−0.08	0.04	−0.26	−0.03		
10. Age	0.07	−0.09	0.04	−0.03	−0.13	−0.08	0.09	−0.06	0.03	
11. Socio-economic status	0.11	0.22	0.19	0.09	0.27	0.06	0.16	0.14	0.02	−0.10

Note. Pearson correlation analysis is used for correlations between continuous variables, and Point-biserial is used for correlations with gender (Boys = 1, Girls = 2). Higher scores on scales represent more positive values (higher general self-efficacy, better relationships, more participation, feeling safer, more access, higher socio-economic status).

**Table 4 behavsci-15-01484-t004:** Hierarchical Regression Analysis for the Prediction of General Self-Efficacy (*n* = 15,040).

	Model 1 (*R*^2^ = 0.164)		Model 2 (*R*^2^ = 0.196)		Model 3 (*R*^2^ = 0.200)	
Variable	*β*	95% *CI* of *β*	*β*	95% *CI* of *β*	*β*	95% *CI* of *β*
Parental involvement	−0.02	[−0.04, −0.01]	0.02	[0.00, 0.03]	0.07	[−0.03, 0.17]
Relation to peers	0.21	[0.19, 0.23]	0.20	[0.18, 0.22]	0.09	[−0.03, 0.21]
Academic and social relation to teachers	0.14	[0.12, 0.16]	0.13	[0.11, 0.14]	0.05	[−0.07, 0.16]
Participation in physical activities	0.08	[0.07, 0.10]	0.07	[0.06, 0.09]	0.16	[0.05, 0.26]
Participation in organized music/cultural leisure activities	0.02	[0.01, 0.04]	0.03	[0.02, 0.05]	0.02	[−0.08, 0.12]
Perceived safety in the neighbourhood	0.14	[0.12, 0.16]	0.08	[0.06, 0.10]	0.16	[0.05, 0.27]
Perceived access to arenas for physical and social activities in the neighbourhood	−0.02	[−0.04, −0.01]	−0.02	[−0.03, 0.00]	0.01	[−0.09, 0.11]
Gender			−0.17	[−0.19, −0.16]	−0.17	[−0.19, −0.16]
Age			0.08	[0.07, 0.10]	0.08	[0.06, 0.09]
Socio-economic status			0.04	[0.02, 0.06]	0.04	[0.03, 0.06]
Gender × Parental involvement					−0.01	[−0.06, 0.04]
Gender × Relation to peers					0.07	[0.01, 0.13]
Gender × Academic and social relation to teachers					0.11	[0.06, 0.17]
Gender × Participation in physical activities					−0.03	[−0.08, 0.02]
Gender × Participation in organized music/cultural leisure activities					0.00	[−0.05, 0.05]
Gender × Perceived safety in the neighbourhood					−0.10	[−0.16, −0.04]
Gender × Perceived access to arenas for physical and social activities in the neighbourhood					−0.01	[−0.06, 0.04]
Age × Parental involvement					−0.10	[−0.14, −0.06]
Age × Relation to peers					−0.04	[−0.08, 0.01]
Age × Academic and social relation to teachers					−0.01	[−0.05, 0.03]
Age × Participation in physical activities					−0.01	[−0.05, 0.03]
Age × Participation in organized music/cultural leisure activities					0.00	[−0.04, 0.03]
Age × Perceived safety in the neighbourhood					0.03	[−0.01, 0.07]
Age × Perceived access to arenas for physical and social activities in the neighbourhood					0.01	[−0.02, 0.05]
Socio-economic status × Parental involvement					0.05	[−0.03, 0.13]
Socio-economic status × Relation to peers					0.08	[−0.02, 0.17]
Socio-economic status × Academic and social relation to teachers					−0.02	[−0.11, 0.07]
Socio-economic status × Participation in physical activities					−0.05	[−0.14, 0.04]
Socio-economic status × Participation in organized music/cultural leisure activities					0.01	[−0.07, 0.09]
Socio-economic status × Perceived safety in the neighbourhood					−0.01	[−0.09, 0.08]
Socio-economic status × Perceived access to arenas for physical and social activities in the neighbourhood					−0.04	[−0.12, 0.05]

Note. Gender: Boys = 1, Girls = 2. Significant effects are marked in bold. Abbreviations: *R*^2^ = Explained variance, *β* = Standardized Beta, *CI* = Confidence Interval. Model 1 df = (7, 15,032), Model 2 df = (10, 15,029), Model 3 df = (31, 15,008).

## Data Availability

The data that support the findings of this study are available for researchers on request to the Norwegian Agency for Shared Services in Education and Research ([89]).

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
