# Peer review of "Environmental and Socio-Demographic Influences on General Self-Efficacy in Norwegian Adolescents"

_behavsci, 2025, doi:10.3390/bs15111484_

Round 1
Reviewer 1 Report
Comments and Suggestions for Authors
Overall Recommendation: Accept after minor revision
The study provides valuable insights into adolescent self-efficacy determinants but would benefit from addressing some interpretive and methodological considerations before publication.
Comments and Suggestions for Authors
This manuscript presents a well-designed cross-sectional study examining environmental and socio-demographic predictors of general self-efficacy among Norwegian adolescents. The work addresses an important public health concern using a large, population-based sample and sophisticated analytical approaches. While the overall quality is high, several areas warrant attention to strengthen the contribution.
Your comprehensive approach to examining multiple environmental contexts simultaneously represents a significant strength. The use of exploratory factor analysis to derive meaningful constructs from the 27 initial items demonstrates methodological rigor, and the resulting seven-factor solution provides a coherent framework for understanding environmental influences on self-efficacy. The hierarchical regression approach effectively demonstrates the incremental contributions of environmental and socio-demographic factors while revealing important interaction effects.
However, the substantial proportion of excluded participants (30% due to missing data) raises concerns about potential selection bias. You acknowledge that excluded participants differed on key variables (gender, age, socioeconomic status), but the implications for generalizability deserve more thorough discussion. Consider conducting sensitivity analyses or describing the missing data patterns in greater detail to help readers assess the robustness of your findings.
The timing of data collection during the COVID-19 pandemic presents both challenges and opportunities that require more careful consideration. While you acknowledge potential pandemic effects on both predictors and outcomes, the discussion could be expanded to consider how this unique context might actually strengthen certain findings. For example, if peer relationships remained predictive of self-efficacy even during periods of social restriction, this might suggest particularly robust associations.
Your theoretical framework effectively integrates self-efficacy theory with socio-ecological models, but the discussion of mechanisms could be strengthened. While you identify which environmental factors predict self-efficacy, deeper exploration of why these relationships exist would enhance the theoretical contribution. For instance, the differential gender effects in the associations between neighborhood safety and self-efficacy deserve more nuanced theoretical interpretation beyond the brief discussion of gender roles and physical capabilities.
The practical implications section is commendable in its specificity, but some recommendations extend beyond what your cross-sectional data can support. Statements about causality and intervention design should be more carefully qualified. Consider framing recommendations as hypotheses for intervention development rather than definitive guidance.
Your finding that parental involvement shows age-dependent effects is particularly interesting and deserves more theoretical development. The interaction suggests that parenting approaches optimal for younger adolescents may become counterproductive for older teens, which has important implications for both theory and practice. This finding could be better integrated with developmental theories of autonomy and attachment.
The modest explained variance (20%) in your final model, while typical for social science research, suggests important predictors remain unidentified. Your discussion of this limitation is appropriate, but consider expanding the theoretical discussion of what these missing factors might include, perhaps drawing on recent developments in positive psychology or resilience research.
Some technical aspects require minor attention. The factor analysis results are well-presented, but providing fit indices (CFI, RMSEA, SRMR) would strengthen the psychometric evaluation. Additionally, while you report confidence intervals for regression coefficients, effect size interpretations could be more consistently provided throughout the results section.
This manuscript represents a solid empirical contribution to understanding adolescent self-efficacy development within a socio-ecological framework. The authors employ appropriate methodology with a substantial sample size and address an important public health concern. The work is well-written and demonstrates good integration of theory and empirical findings.
The main strengths include the comprehensive examination of multiple environmental contexts, sophisticated statistical analysis with proper attention to interaction effects, and practical implications grounded in established theory. The large Norwegian sample provides valuable cultural context that extends beyond typical North American research populations.
The primary limitations involve the cross-sectional design constraining causal inferences, substantial missing data affecting generalizability, and the unique pandemic context requiring careful interpretation. However, the authors appropriately acknowledge these limitations and discuss their implications.
The manuscript would benefit from minor revisions addressing the missing data patterns more thoroughly, expanding the theoretical discussion of mechanisms, and providing more careful qualification of practical recommendations. With these revisions, this work would make a meaningful contribution to the adolescent development and public health literature.
The study's focus on positive mental health constructs and multi-level intervention implications aligns well with current priorities in adolescent health promotion. The findings provide actionable insights for educators, parents, and policymakers working to support adolescent well-being.
Author Response
COMMENTS REVIEWER 1
Overall Recommendation: Accept after minor revision
The study provides valuable insights into adolescent self-efficacy determinants but would benefit from addressing some interpretive and methodological considerations before publication.
Comments and Suggestions for Authors
This manuscript presents a well-designed cross-sectional study examining environmental and socio-demographic predictors of general self-efficacy among Norwegian adolescents. The work addresses an important public health concern using a large, population-based sample and sophisticated analytical approaches. While the overall quality is high, several areas warrant attention to strengthen the contribution.
Your comprehensive approach to examining multiple environmental contexts simultaneously represents a significant strength. The use of exploratory factor analysis to derive meaningful constructs from the 27 initial items demonstrates methodological rigor, and the resulting seven-factor solution provides a coherent framework for understanding environmental influences on self-efficacy. The hierarchical regression approach effectively demonstrates the incremental contributions of environmental and socio-demographic factors while revealing important interaction effects.
However, the substantial proportion of excluded participants (30% due to missing data) raises concerns about potential selection bias. You acknowledge that excluded participants differed on key variables (gender, age, socioeconomic status), but the implications for generalizability deserve more thorough discussion. Consider conducting sensitivity analyses or describing the missing data patterns in greater detail to help readers assess the robustness of your findings.
Response: We have now conducted an MCAR test, which was significant (X2(183)=1663,611, p= .000), indicating that the data are not missing completely at random. To assess the potential impact of missing data, sensitivity analyses were conducted using the Expectation-Maximization (EM) algorithm in SPSS to impute missing data (excluding gender, as EM does not support categorical variable imputation, although gender had only 2% missing data). We reanalyzed our regression model using this imputed dataset. The results were largely consistent with the original analysis. We have added information about this in the Limitation section where the robustness of our findings is addressed (see lines 665-671 in clean manuscript version). Results from the sensitivity analysis are included in the Supplementary Table 1S.
The timing of data collection during the COVID-19 pandemic presents both challenges and opportunities that require more careful consideration. While you acknowledge potential pandemic effects on both predictors and outcomes, the discussion could be expanded to consider how this unique context might actually strengthen certain findings. For example, if peer relationships remained predictive of self-efficacy even during periods of social restriction, this might suggest particularly robust associations.
Thank you for this insightful feedback. We appreciate your suggestion and have now expanded the discussion to include the possibility that the unique context of the Covid-19 pandemic may have strengthened certain findings (see lines 658-660).
Your theoretical framework effectively integrates self-efficacy theory with socio-ecological models, but the discussion of mechanisms could be strengthened. While you identify which environmental factors predict self-efficacy, deeper exploration of why these relationships exist would enhance the theoretical contribution. For instance, the differential gender effects in the associations between neighborhood safety and self-efficacy deserve more nuanced theoretical interpretation beyond the brief discussion of gender roles and physical capabilities.
Thank you for this comment. We agree that the associations between environmental factors and GSE warranted a deeper discussion of potential underlying mechanisms. We have now expanded the discussion section accordingly, including a more nuanced interpretation of the gender x neighbourhood safety interaction effect (e.g., see lines 435-440 and 509-524).
The practical implications section is commendable in its specificity, but some recommendations extend beyond what your cross-sectional data can support. Statements about causality and intervention design should be more carefully qualified. Consider framing recommendations as hypotheses for intervention development rather than definitive guidance.
Thank you for this valuable feedback. We have revised the practical implications section to include more cautious language. This should better align the recommendations with the limitations of the cross-sectional data.
Your finding that parental involvement shows age-dependent effects is particularly interesting and deserves more theoretical development. The interaction suggests that parenting approaches optimal for younger adolescents may become counterproductive for older teens, which has important implications for both theory and practice. This finding could be better integrated with developmental theories of autonomy and attachment.
Thank you for this feedback. We have expanded the discussion to further develop this finding in the context of developmental theories of autonomy and attachment (see lines 460-476).
The modest explained variance (20%) in your final model, while typical for social science research, suggests important predictors remain unidentified. Your discussion of this limitation is appropriate, but consider expanding the theoretical discussion of what these missing factors might include, perhaps drawing on recent developments in positive psychology or resilience research.
Thank you for this relevant feedback. We have expanded our discussion to include additional factors that could be considered in a more comprehensive model for explaining adolescents’ general self-efficacy. Specifically, we have highlighted the potential contributions of resilience-related factors such as emotion regulation and hardiness (see lines 642-645).
Some technical aspects require minor attention. The factor analysis results are well-presented, but providing fit indices (CFI, RMSEA, SRMR) would strengthen the psychometric evaluation.
Thank you for this feedback. As this study employed exploratory factor analysis (EFA) using SPSS, the software does not provide these specific fit indices, which are generally used in confirmatory factor analysis (CFA).
We ensured the robustness of our factor analysis by thoroughly examining key indicators such as eigenvalues, the scree plot, factor loadings, and explained variance. The total explained variance has now been added to Table 1. We believe this approach is appropriate for the exploratory nature of the current factor analysis.
Additionally, while you report confidence intervals for regression coefficients, effect size interpretations could be more consistently provided throughout the results section.
We have now provided interpretation of the effect sizes where this was not included.
This manuscript represents a solid empirical contribution to understanding adolescent self-efficacy development within a socio-ecological framework. The authors employ appropriate methodology with a substantial sample size and address an important public health concern. The work is well-written and demonstrates good integration of theory and empirical findings.
The main strengths include the comprehensive examination of multiple environmental contexts, sophisticated statistical analysis with proper attention to interaction effects, and practical implications grounded in established theory. The large Norwegian sample provides valuable cultural context that extends beyond typical North American research populations.
The primary limitations involve the cross-sectional design constraining causal inferences, substantial missing data affecting generalizability, and the unique pandemic context requiring careful interpretation. However, the authors appropriately acknowledge these limitations and discuss their implications.
The manuscript would benefit from minor revisions addressing the missing data patterns more thoroughly, expanding the theoretical discussion of mechanisms, and providing more careful qualification of practical recommendations. With these revisions, this work would make a meaningful contribution to the adolescent development and public health literature.
The study's focus on positive mental health constructs and multi-level intervention implications aligns well with current priorities in adolescent health promotion. The findings provide actionable insights for educators, parents, and policymakers working to support adolescent well-being.
Thank you for this feedback!
Reviewer 2 Report
Comments and Suggestions for Authors
Thank you for the opportunity to review this interesting manuscript. I find the topic timely, and the research questions grounded in prior work. A clear strength of the study is the large sample size and the breadth of measures included.
In this study, the environmental factors appear to have been chosen based on what was available in the dataset. While these align with theory, this nonetheless represents a somewhat post hoc solution. In the introduction the authors report on prior literature, and based on the general impression, hypotheses could be made based on prior literature. Instead, the authors choose an explorative approach. I think the storyline could be strengthened by including hypotheses, at least when possible. Also, I find the storyline lacking a description of general vs specific self-efficacy, as well as the influence of situated emotional and physiological states on self-efficacy. The reference list seems adequate, but since self-efficacy is not my main area of research, I hope the other reviewers can comment on the references, and also that the literature is adequately presented.
I appreciate that the authors conducted a factor analysis to identify adequate environmental factors. Also, I find the description of the dataset and procedures clear, with the exception of information on ethical permission. As I understand it, it was not sought for or provided by an Ethical Board. Can you clarify this, and the common ethical procedures in Norway?
Data handling and analyses seem adequate, and presentation likewise.
The associations reported between general self-efficacy (GSE) and environmental factors are both interesting and valuable. At the same time, these associations are studied within a cross-sectional design. Specifically, it is not possible to establish directionality: supportive environments may foster higher GSE, but it is also plausible that adolescents with higher GSE are better able to create or perceive supportive environments. Moreover, some of the included variables may be considered indicators of broader psychosocial wellbeing or quality of life, raising the possibility of confounding or overlapping constructs. It would strengthen the discussion to situate the findings within a broader framework of adolescent wellbeing. Also, I suggest avoiding causal language, instead of using “predictor” (even if statistically accurate), it would be preferable to frame the results in terms of associations. Also, it could be informative to include a short description of the Norwegian context (wellbeing, schooling, society), espcially during the covid-19 pandemic.
I appreciate the way the limitations of the study are discussed, as well as the recommendation to pursue longitudinal research. However, the manuscript mentions for the first time in the limitations section that the data were collected during the pandemic. This may be a significant factor influencing mean levels of many of the included variables. Would it be possible to compare mean levels (e.g., of safety) with pre- or post-COVID data, to provide a general picture of how the pandemic context may have affected the findings? Also, as mentioned above, I suggest the authors add a description about the societal situation in the introduction.
Regarding the section on gender roles (page 20, beginning), I would add to the discussion: It is also likely that boys face a higher risk of physical violence than girls, adding to the felt unsafety. The explanation provided, highlighting the risk of harassment faced by girls, is valid also. Certain subgroups of boys may be particularly vulnerable to physical violence, and this deserves acknowledgment.
Finally, the manuscript refers at times to “modifiable predictors.” I find this wording potentially misleading. For instance, many environmental aspects are not under adolescents’ control. Living in a disadvantaged neighborhood, for example, may increase risk behaviors and exposure to vicarious experiences of non-mastery. I aprreciate the inclusion of suggestions for system-level interventions.
Author Response
COMMENTS REVIEWER 2
Comments and Suggestions for Authors
Thank you for the opportunity to review this interesting manuscript. I find the topic timely, and the research questions grounded in prior work. A clear strength of the study is the large sample size and the breadth of measures included.
In this study, the environmental factors appear to have been chosen based on what was available in the dataset. While these align with theory, this nonetheless represents a somewhat post hoc solution. In the introduction the authors report on prior literature, and based on the general impression, hypotheses could be made based on prior literature. Instead, the authors choose an explorative approach. I think the storyline could be strengthened by including hypotheses, at least when possible.
Response: Thank you for this feedback. We have now included hypotheses related to the two first aims (see lines 176-185 in clean manuscript version).
Also, I find the storyline lacking a description of general vs specific self-efficacy, as well as the influence of situated emotional and physiological states on self-efficacy.
Thank you for these comments. A brief description of general vs. specific self-efficacy has now been included (see lines 47-50), as well as the role of situated emotional and physiological states on self-efficacy (see lines 83-84).
The reference list seems adequate, but since self-efficacy is not my main area of research, I hope the other reviewers can comment on the references, and also that the literature is adequately presented.
I appreciate that the authors conducted a factor analysis to identify adequate environmental factors. Also, I find the description of the dataset and procedures clear, with the exception of information on ethical permission. As I understand it, it was not sought for or provided by an Ethical Board. Can you clarify this, and the common ethical procedures in Norway?
Thank you for raising this point. Access to the data for this study was provided by the Norwegian Social Research Institute (NOVA), which coordinates the Ungdata project. The data used in this study were pre-existing and fully anonymized before we were granted access. NOVA obtained ethical approval for data collection from the Norwegian Agency for Shared Services in Education and Research (Sikt). Sikt provides data protection services to 130 Norwegian research and education institutions, evaluating the planned processing of personal data to ensure compliance with data protection regulations.
We recognize that this information was not adequately detailed in Chapter 2.5 of the manuscript, and we have now revised this section to improve clarity (see lines 324-328).
Data handling and analyses seem adequate, and presentation likewise.
Thanks for this feedback.
The associations reported between general self-efficacy (GSE) and environmental factors are both interesting and valuable. At the same time, these associations are studied within a cross-sectional design. Specifically, it is not possible to establish directionality: supportive environments may foster higher GSE, but it is also plausible that adolescents with higher GSE are better able to create or perceive supportive environments. Moreover, some of the included variables may be considered indicators of broader psychosocial wellbeing or quality of life, raising the possibility of confounding or overlapping constructs. It would strengthen the discussion to situate the findings within a broader framework of adolescent wellbeing.
Thank you for this valuable feedback. The limitation of the cross-sectional design in establishing causality and directionality was already addressed in the Limitations section. However, we have now expanded this discussion by situating our findings within a broader framework of adolescent well-being and incorporating your points regarding potential reciprocal relationships, confounding effects, and the need to assess overlapping constructs (see lines 625-636).
Also, I suggest avoiding causal language, instead of using “predictor” (even if statistically accurate), it would be preferable to frame the results in terms of associations
Thank you for this feedback. We have revised the discussion section in line with this suggestion.
Also, it could be informative to include a short description of the Norwegian context (wellbeing, schooling, society), especially during the covid-19 pandemic.
Thank you for this valuable feedback. We have expanded the Methods section to include a brief description of the Norwegian context at the time of data collection (see lines 216-220).
I appreciate the way the limitations of the study are discussed, as well as the recommendation to pursue longitudinal research. However, the manuscript mentions for the first time in the limitations section that the data were collected during the pandemic. This may be a significant factor influencing mean levels of many of the included variables. Would it be possible to compare mean levels (e.g., of safety) with pre- or post-COVID data, to provide a general picture of how the pandemic context may have affected the findings?
Thank you for this suggestion. We agree that comparing changes in certain variables from pre- to post-COVID would provide valuable insights into how the pandemic context may have influenced our findings. Unfortunately, we do not have readily access to longitudinal data on these variables, and conducting such analyses would also fall beyond the scope of this study. However, we have now incorporated findings from a study examining changes in participation in organized leisure activities between pre-pandemic and pandemic periods into the discussion in the Limitations section (see lines 652-655).
Also, as mentioned above, I suggest the authors add a description about the societal situation in the introduction.
Information about the societal context in Norway at the time of data collection and the preceding period, has now been included. However, we found it more appropriate to place this information in the Methods section, alongside information about the sample (see lines 216-220).
Regarding the section on gender roles (page 20, beginning), I would add to the discussion: It is also likely that boys face a higher risk of physical violence than girls, adding to the felt unsafety. The explanation provided, highlighting the risk of harassment faced by girls, is valid also. Certain subgroups of boys may be particularly vulnerable to physical violence, and this deserves acknowledgment.
Thank you for this valuable feedback. We have expanded the discussion on potential explanations for the gender x neighbourhood safety interaction effect on general self-efficacy. This now includes a focus on gendered differences in neighbourhood exposure, including negative effects such as violence (see lines 509-524).
Finally, the manuscript refers at times to “modifiable predictors.” I find this wording potentially misleading. For instance, many environmental aspects are not under adolescents’ control. Living in a disadvantaged neighborhood, for example, may increase risk behaviors and exposure to vicarious experiences of non-mastery.
Thank you for this feedback. We agree that many environmental conditions, including neighbourhood factors, are not directly under adolescents’ control. However, when we refer to these conditions as “modifiable”, we mean that they can be influenced through interventions and efforts at higher socio-ecological levels, such as community programs, policy changes, or school-based initiatives. Such a focus is embedded in our discussion of practical implications.
I appreciate the inclusion of suggestions for system-level interventions.
Thank you for this feedback.
Reviewer 3 Report
Comments and Suggestions for Authors
ARTICLE REVIEW COMMENTS Environmental and Socio-Demographic Influences on General Self-Efficacy in Norwegian Adolescents
We would like to thank the authors for their study, as this is a topic that, as they mention in their article, has been little researched.
The abstract is well structured and covers the most relevant aspects of the study.
The introduction is well structured, moving from the general to the specific in relation to theoretical and research aspects of the variables used in the study, reviewing previous studies on the subject of the article. The introduction concludes with the objectives, which are well stated.
In line with the above, neither the abstract nor the introduction require any changes or modifications.
Method
The section on methodology adequately covers the relevant aspects of the study. However, I would like to share a brief reflection with the authors, which does not necessarily need to be addressed in the study, as it is only a suggestion. This suggestion refers to the counties of Vestfold and Telemark in south-eastern Norway, where specific aspects that may differ from other Norwegian counties could be described, such as unemployment rates, income levels, crime rates, etc.
Results
These are well presented and described in tables and in the text. No changes are required in this section.
Discussion and conclusions
These sections are well developed, covering the most relevant aspects of the study, relating the results to other research, and including a section on practical implications, limitations and future studies. In view of this, no changes are required in this section.
References
Note that the last two references are not in alphabetical order.
Author Response
COMMENTS REVIEWER 3
Comments and Suggestions for Authors
ARTICLE REVIEW COMMENTS Environmental and Socio-Demographic Influences on General Self-Efficacy in Norwegian Adolescents
We would like to thank the authors for their study, as this is a topic that, as they mention in their article, has been little researched.
The abstract is well structured and covers the most relevant aspects of the study.
The introduction is well structured, moving from the general to the specific in relation to theoretical and research aspects of the variables used in the study, reviewing previous studies on the subject of the article. The introduction concludes with the objectives, which are well stated.
In line with the above, neither the abstract nor the introduction require any changes or modifications.
Thank you for this feedback.
Method
The section on methodology adequately covers the relevant aspects of the study. However, I would like to share a brief reflection with the authors, which does not necessarily need to be addressed in the study, as it is only a suggestion. This suggestion refers to the counties of Vestfold and Telemark in south-eastern Norway, where specific aspects that may differ from other Norwegian counties could be described, such as unemployment rates, income levels, crime rates, etc.
Response: Thank you for suggesting the inclusion of contextual factors relevant to the generalizability of our findings. We have now incorporated a discussion of relevant differences in the Limitations section, specifically addressing the generalizability of our results (see lines 675-678).
Results
These are well presented and described in tables and in the text. No changes are required in this section.
Thank you for this feedback.
Discussion and conclusions
These sections are well developed, covering the most relevant aspects of the study, relating the results to other research, and including a section on practical implications, limitations and future studies. In view of this, no changes are required in this section.
Thank you for this feedback.
References
Note that the last two references are not in alphabetical order.
Thank you for the reminder. This is related to the Norwegian alphabet and the settings in the reference management software we are using. We will ensure that the references are correctly ordered before final publication.
Round 2
Reviewer 2 Report
Comments and Suggestions for Authors
The authors have adequately addressed all my comments. I recommend the ms. for publication.